# AIBP: A New Safeguard against Glaucomatous Neuroinflammation

**DOI:** 10.3390/cells13020198

**Published:** 2024-01-21

**Authors:** Seunghwan Choi, Soo-Ho Choi, Tonking Bastola, Younggun Park, Jonghyun Oh, Keun-Young Kim, Sinwoo Hwang, Yury I. Miller, Won-Kyu Ju

**Affiliations:** 1Hamilton Glaucoma Center and Shiley Eye Institute, Viterbi Family Department of Ophthalmology, University of California San Diego, La Jolla, CA 92093, USA; sec009@health.ucsd.edu (S.C.); tbastola@health.ucsd.edu (T.B.); cuteyg2000@gmail.com (Y.P.);; 2Department of Medicine, University of California San Diego, La Jolla, CA 92093, USA; 3Department of Ophthalmology and Visual Science, Seoul St. Mary’s Hospital, College of Medicine, The Catholic University of Korea, Seoul 06591, Republic of Korea; 4Department of Ophthalmology, Dongguk University Ilsan Hospital, Goyang 10326, Republic of Korea; 5National Center for Microscopy and Imaging Research, Department of Neurosciences, University of California San Diego, La Jolla, CA 92093, USA

**Keywords:** AIBP, glaucoma, neuroinflammation, cholesterol, TLR4, lipid rafts, mitochondria, gene therapy, AAV

## Abstract

Glaucoma is a group of ocular diseases that cause irreversible blindness. It is characterized by multifactorial degeneration of the optic nerve axons and retinal ganglion cells (RGCs), resulting in the loss of vision. Major components of glaucoma pathogenesis include glia-driven neuroinflammation and impairment of mitochondrial dynamics and bioenergetics, leading to retinal neurodegeneration. In this review article, we summarize current evidence for the emerging role of apolipoprotein A-I binding protein (AIBP) as an important anti-inflammatory and neuroprotective factor in the retina. Due to its association with toll-like receptor 4 (TLR4), extracellular AIBP selectively removes excess cholesterol from the plasma membrane of inflammatory and activated cells. This results in the reduced expression of TLR4-associated, cholesterol-rich lipid rafts and the inhibition of downstream inflammatory signaling. Intracellular AIBP is localized to mitochondria and modulates mitophagy through the ubiquitination of mitofusins 1 and 2. Importantly, elevated intraocular pressure induces AIBP deficiency in mouse models and in human glaucomatous retina. AIBP deficiency leads to the activation of TLR4 in Müller glia, triggering mitochondrial dysfunction in both RGCs and Müller glia, and compromising visual function in a mouse model. Conversely, restoring AIBP expression in the retina reduces neuroinflammation, prevents RGCs death, and protects visual function. These results provide new insight into the mechanism of AIBP function in the retina and suggest a therapeutic potential for restoring retinal AIBP expression in the treatment of glaucoma.

## 1. Introduction

Glaucoma is a leading cause of irreversible blindness globally. It is characterized by a slow, progressive, and irreversible degeneration of retinal ganglion cells (RGCs) and their axons, resulting in a cupping of the optic disc (also called the optic nerve head) and visual field loss in patients with glaucoma [1,2,3]. It has been reported that the global prevalence of glaucoma is currently 3.5% of the population aged 40–80 years, and the number of glaucoma patients is expected to reach over 111.8 million by 2040 [4].

Glaucoma is a multifactorial neurodegenerative disorder that is caused by multiple pathogenic stressors, including elevated intraocular pressure (IOP), aging, oxidative/metabolic stress, mitochondrial dysfunction, defects in the retinal vasculature, genetic factors, and neuroinflammation [1,5,6]. Despite an increasing number of publications from animal models and human subjects, the underlying pathophysiological mechanisms of glaucoma are not fully understood. To date, IOP is only one proven, modifiable, and treatable risk factor, and lowering IOP pharmaceutically or surgically can reduce glaucoma progression [2,3,7]. However, the reduction of IOP in some patients is often insufficient to restore visual function [8], and the precise mechanisms by which increased IOP leads to axonal degeneration and RGC death remain to be elucidated. The mechanical compression of the axons induced by high IOP at the level of the lamina cribrosa might directly lead to ischemia-hypoxia damage, impairment of axonal transport, deprivation of growth factors, and subsequent RGC loss [9,10,11]. In addition to IOP elevation, accumulating evidence suggested that various mechanisms, such as oxidative stress, mitochondrial dysfunction, and inflammation, contribute to glaucomatous neurodegeneration. Furthermore, crosstalk between mitochondrial dysfunction, inflammasome activation, and inflammatory response exacerbates the progression of glaucoma [9,12,13].

Apolipoprotein A-I (APOA-I) binding protein (AIBP, gene name *APOA1BP*, also known as *NAXE*) was identified as a binding partner of human APOA-I, which is the major component of high-density lipoprotein (HDL) [14]. The human *APOA1BP* gene is located on chromosome 1q21, and AIBP expression is ubiquitous and constitutive in human and mouse tissues, including the kidney, brain, liver, thyroid/adrenal glands, testis, and retina [14]. Secreted AIBP promotes cholesterol efflux to HDL/APOA-I in endothelial cells, macrophages, and microglia, resulting in increased cholesterol depletion from the plasma membrane of inflamed cells [15,16,17,18,19]. AIBP protein can be found in the plasma of sepsis patients and healthy subject’s urine and cerebrospinal fluid [14]. In the cell culture system, lipopolysaccharides (LPS) and oxidized LDL increase AIBP expression at the translational level in macrophages [18,20].

In systemic naïve AIBP knockout (*Apoa1bp^−/−^*) mice, AIBP deficiency triggered glia-driven neuroinflammation and induced mitochondrial dysfunction in both RGCs and Müller glial cells, leading to visual dysfunction [12]. Conversely, administration of recombinant AIBP prevented Müller glia-mediated inflammatory response and promoted RGC survival in a mouse model of acute IOP elevation. Intriguingly, AIBP protein expression is reduced in glaucomatous human and mouse retinas [12,13]. Based on these results, we propose that restoring AIBP expression in the retina can provide neuroprotection in glaucoma.

Since studies in humans and mouse models have shown that AIBP regulates lipid raft-induced inflammation and cholesterol metabolism in multiple cell types [12,15,17,19,20,21,22], a comprehensive understanding of AIBP-regulated cholesterol metabolism, mitochondrial function, and inflammation involved in glaucoma progression is necessary to develop an attractive therapy. This article will review recent findings related to the biological functions of AIBP in the context of glaucoma.

## 2. Potential Role of AIBP in Lipid Rafts and Cholesterol Metabolism in Glaucomatous Retina

Given that lowering IOP is insufficient to prevent the disease progression in some patients with glaucoma, one of the significant potential risk factors influencing glaucoma progression is neuroinflammation associated with toll-like receptor 4 (TLR4) activation, cholesterol dysregulation, inflammasome activation, and inflammatory responses [5]. Cholesterol is a necessary component of cell structure. However, excessive cellular accumulation of cholesterol is toxic and exerts detrimental effects on cellular functions. It is well appreciated that cholesterol accumulation in immune cells can promote TLR signaling and inflammasome activations, leading to inflammation in the context of metabolic diseases [23]. In neurodegenerative diseases, e.g., Alzheimer’s disease, a high cholesterol level has been reported as a risk factor for disease progression [24]. Thus, cholesterol has emerged as a regulator of diverse cellular metabolisms and signaling pathways.

A recent review extensively discussed lipid raft-associated lipid metabolism in the pathophysiology of retinal diseases [25]. Lipid rafts are dynamic microdomains in the plasma membrane that act as platforms for intracellular signaling pathways involved in cell migration, membrane trafficking, and inflammation [25]. Cholesterol is a major component of lipid rafts, and its depletion disrupts lipid rafts, leading to the inhibition of receptor-mediated cellular signaling. The new concept of an inflammaraft, a pathological lipid raft, was recently introduced to understand the function of enlarged lipid rafts with concentrated inflammatory receptors and adaptor proteins, such as TLR4, in inflamed cells [26]. Based on previous studies, we characterized the inflammarafts with increased lipid raft abundance in inflammatory cells, increased cholesterol levels, and prolonged inflammatory receptors occupancy in lipid rafts [19,22,26]. In macrophages and microglia, which highly express TLR4 on the plasma membrane, LPS leads to activation of the inflammatory response, increasing of lipid raft abundance, and ensuing increase of TLR4 recruitment to lipid rafts via modulation of ABCA1-dependent cholesterol efflux pathway [27,28].

Epidemiology studies have revealed that single nucleotide polymorphisms (SNPs) in lipid metabolism-related genes are linked to the progression of primary open-angle glaucoma (POAG) [29,30,31,32]. In particular, SNPs within the gene responsible for encoding ATP-binding cassette transporter A1 (ABCA1), a membrane transporter involved in phospholipids and cholesterol, have been reported to be linked with human glaucoma [29,33,34,35]. ABCA1 deficiency in the retinal astrocytes induced glaucoma-like optic neuropathy in aged mice and resulted in RGC degeneration [36]. Furthermore, increased ABCA1 expression reduced retinal degeneration and prevented RGC death [37]. Of interest, our recent and ongoing studies have revealed that deficiencies of AIBP and ABCA1, crucial regulators of cholesterol metabolism, are notably evident in RGCs and Müller glia endfeet in both glaucomatous human and mouse retinas. These results link to the degeneration of RGCs [12,13]. Importantly, these findings strongly suggest that dysregulation of cholesterol homeostasis plays a critical role in RGC degeneration and glial dysfunction in glaucomatous neuroinflammation and neurodegeneration.

Interestingly, findings of the effect of AIBP on regulating cholesterol efflux show that AIBP increases ABCA1 stability and prevents ubiquitin-mediated degradation via facilitating ABCA1-APOA1 binding in macrophages [38]. Moreover, the combination of AIBP/APOA1 and anti-VEGF treatment showed a beneficial effect on overcoming anti-VEGF resistance via facilitating cholesterol efflux from lipid-laden macrophages in a mouse model of laser-induced choroidal neovascularization [39]. Regarding cancer biology, the anti-tumor effects of AIBP are associated with the regulation of cholesterol efflux and reactive oxygen species (ROS) scavenging in intestinal tumors and hepatocellular carcinoma, respectively [40,41]. Our recent studies demonstrated that AIBP enhanced cholesterol efflux from TLR4-associated lipid rafts and subsequently reduced TLR4-mediated inflammatory response in activated macrophages and microglia [16,17,18,19]. Remarkably, a single intrathecal injection of AIBP in a mouse model of neuropathic pain showed a long-lasting therapeutic effect and induced cholesterol metabolism reprograming in spinal microglia [19], implying that cholesterol-driven TLR4-associated lipid raft formation engages in the development of neuropathic pain.

AIBP and ABCA1 deficiency is linked to increased expression of TLR4/IL-1β and MAPKs, which contributes to RGC degeneration in response to elevated IOP in the retina of glaucomatous DBA/2J mice [12]. However, the pathophysiological mechanisms of cholesterol efflux in glaucoma progression and how AIBP regulates retinal cholesterol metabolism and TLR4-associated lipid raft formation remain to be explored. Given that the proposed mechanism for AIBP-mediated RGC protection is associated with cholesterol efflux from inflamed retinal cells, a critical question of great significance is which cholesterol acceptors in the retina serve as a key player in regulating the elimination of excessive cholesterol through the AIBP-ABCA1 pathway. In the brain, which shares many characteristics with the retina and optic nerve, cholesterol fully relies on local biosynthesis because the blood–brain barrier (BBB) is not permeable to circulating blood lipoproteins [42,43]. Astrocytes are major sources of cholesterol synthesis and provide newly synthesized cholesterol to neurons in the brain [44]. APOE-containing HDL-like lipoproteins produced by astrocytes are involved in cholesterol trafficking to neurons via ABCG1/G4 receptors in the brain [25]. *APOA1* gene is not expressed in the brain [44], but APOA1 protein is specifically transported from the periphery to the brain [45,46]. Unlike the BBB, the blood–retina barrier is highly permeable to transfer lipoproteins from plasma to retinal pigment epithelium (RPE) cells and plays a critical role in maintaining retinal homeostasis [25,47]. When plasma lipoproteins enter the retina, lipoprotein receptors such as the LDL receptor, scavenger receptor class B type I/II, and CD36 in RPE cells are involved in lipoprotein uptake [48,49,50,51,52]. Like in the brain, APOE- or APOA1-containing HDL-like lipoproteins regulate cholesterol trafficking in the retina [51,52]. Moreover, retinal APOA1 not only can be synthesized by RPE, but also is delivered from the periphery [25]. Recent clinical evidence demonstrated higher levels of cholesterol and lower levels of HDL in patients with glaucoma and that statins show a protective effect in glaucoma, suggesting that dysregulation of retinal cholesterol metabolism and inflammation may contribute to glaucoma progression [53]. In addition, ABCA1/G1 deficiency increases TLR4-induced proinflammatory gene expression [54,55,56,57], while cholesterol acceptors such as HDL and APOA1 reduce TLR4-dependent inflammatory responses [58,59,60]. Therefore, the facilitation of cholesterol efflux by AIBP to either HDL or APOA1 may play a role in disrupting TLR4-lipid raft formation. This, in turn, may prevent cellular cholesterol accumulation and inhibit inflammatory response in the retina, ultimately mitigating the progression of glaucomatous neuroinflammation (Figure 1).

## 3. Role of AIBP in Regulation of Glial-Driven Retinal Neuroinflammation and RGC Degeneration

Neuroinflammation is associated with the activation of glial cells in the central nervous system, resulting in the release of proinflammatory cytokines and chemokines. Initial inflammation is a protective mechanism that is characterized by repairing damaged tissue and removing cellular debris by microglia and the complement system in the retina. Nevertheless, sustained activation of retinal immune cells contributes to impaired immune function and persistent inflammation, ultimately leading to neuroinflammation in glaucoma. Hence, a comprehensive understanding of the pathophysiological role and therapeutic implication of neuroinflammation is critical to guide the development of potential anti-inflammatory treatments for glaucoma.

Our studies demonstrated that the secreted AIBP protein binds to TLR4, which is a proinflammatory receptor, in activated cells and selectively augments cholesterol efflux from TLR4-associated lipid rafts, leading to the reduction of TLR4 dimerization, anti-inflammatory response, and cholesterol metabolism reprograming in microglia [17,19]. Moreover, a recent report demonstrated that AIBP attenuates paclitaxel-induced transient receptor potential cation channel subfamily V member 1 (TRPV1) activation by reducing TRPV1-associated lipid raft content and TLR4-TRPV1 interaction in dorsal root ganglion neurons in vivo, resulting in alleviation of lipid raft-associated neuropathic pain [22].

TLR4 is one of the pattern recognition receptors (PRRs) that can be activated by dangerous stimuli, including damaged-associated molecular patterns (DAMPs) and pathogen-associated molecular patterns (PAMPs), leading to the secretion of numerous cytokines and chemokines [61,62]. TLR4 can initiate an anti-bacterial response in mammals [63] and is activated in neurodegenerative diseases [64,65,66,67,68]. Studies have shown that activation of TLR4 before exposure to injuries plays a role in neuroprotection [69,70] and increased photoreceptor survival [71]. In contrast, excessive TLR4 activation after injuries induces necroptosis-mediated neuroinflammation and retinal regeneration [72,73,74,75,76]. Proteomic analysis and epidemiological studies of the glaucomatous human retina reveal increased expression levels of TLR2, TLR4, and TLR7, along with their association with SNPs of TLR4 [77,78,79]. Thus, glial activation by TLR4 may contribute to the development of retinal degeneration. Indeed, our recent findings indicate a noteworthy upregulation of TLR4 and IL-1β expression in Müller glia in both glaucomatous human and mouse retinas compared to age/gender-matched control [12]. Activated Müller glia are likely linked to the degeneration of RGCs in patients with glaucoma.

Glial activation in the retina and optic nerve head has been implicated in experimental animal models of glaucoma [80,81,82,83,84,85,86]. In DBA/2J mice, an extensively studied chronic glaucoma model characterized by the spontaneous development of elevated IOP with age [87,88], convincing evidence indicates a strong correlation between alterations in microglia and retinal degeneration [85,89]. Conversely, microglial inactivation showed a protective effect in experimental glaucoma [84,90,91,92]. In other animal models of glaucoma, including rat and feline, the progression of glaucoma is significantly affected by the induction of retinal inflammation through glial activation, increased production of proinflammatory cytokines, and activation of NF-κB and TLRs [93,94,95,96,97]. Previous studies demonstrated that TLR4-dependent signaling activation induced microglial activation and led to RGC loss in experimental glaucoma [98,99]. Optic nerve crush (ONC), an experimental model of traumatic optic neuropathy, induces axonal degeneration and upregulates the release of proinflammatory cytokines and chemokines, resulting in RGC death [100]. It has been reported that TLR4 activation occurs concurrently with the degeneration of RGCs following ONC. This implies that the inflammation induced by TLR4 activation might contribute to the degeneration of RGC death [101,102,103]. Consistent with these findings, the TLR4 inhibitor TAK242 showed similar effects, diminishing astrocyte activation as evidenced by a reduction of glial fibrillary acidic protein expression and subsequently suppressing RGC death following ONC [104]. In mouse models of microbead-induced ocular hypertension and retinal ischemia/reperfusion injury, there were significant increases of microglial activation, expression of proinflammatory cytokines and chemokines, and expression of TLR4 and NLR family pyrin domain containing 3 inflammasome [76,105]. Interestingly, a study from Astafurov and coworkers demonstrated that oral bacteria load was higher in glaucomatous patients and that peripheral administration of LPS into two mouse models of glaucoma, DBA/2J and microbead, increased microglial activation and TLR4 signaling, resulting in subsequent enhancement of axonal degeneration and RGC death [106].

Recent findings from our group showed a significant upregulation of TLR4 and IL-1β in the endfeet of Müller glial cells of the ganglion cell layer in glaucomatous DBA/2J and *Apoa1bp^−/−^* mice [12]. In addition, MAPK signaling pathways, including ERK1/2 and p38, which lead to inflammation, oxidative stress, and mitochondrial dysfunction [107,108,109,110,111,112], are activated in the retina of *Apoa1bp^−/−^* mice and, on the contrary, in vivo recombinant AIBP administration reduces glia-mediated inflammation and RGC death caused by elevated IOP [12]. Considering our findings indicating that AIBP facilitates excess cholesterol removal from the plasma membrane of multiple cell types [17,19,113], AIBP-mediated augmentation of cholesterol efflux from TLR4-associated lipid rafts in inflamed retinal glial cells may contribute to preventing RGC death. However, we have not ruled out the possibility that AIBP may also bind to other receptors and modulate other processes in retinal cells. Collectively, AIBP deficiency potentially plays a role in triggering retinal glial activation and advancing TLR4/IL-1β-induced neuroinflammation, exacerbating glaucoma progression and visual dysfunction.

## 4. Role of AIBP in Mitochondrial Dynamics and Function in Glaucomatous Neuroinflammation and Neurodegeneration

Mitochondria are dynamic organelles that generate adenosine triphosphate (ATP), which is subsequently used as a major source of energy production in the cell, so they are often referred to as the power plants of the cells [114]. While mitochondria are primarily recognized for their role in bioenergy production and metabolite generation, there is increasing attention to their involvement in various other functions, such as cell signaling, programmed cell death, ROS generation and elimination, and neuroinflammation [115].

Given that RGCs are highly susceptible to mitochondrial stress caused by glaucomatous insults such as elevated IOP and oxidative stress [5,6,80,86,116], mitochondrial dysfunction is considered a critical causal factor in glaucomatous neuroinflammation and neurodegeneration [6,12,13]. This could be linked to mitochondrial damage-induced cellular processes, such as aging, excitotoxicity, oxidative stress, inflammasome activation, mitophagy deficiency, and sustained inflammatory response, as downstream pathological pathways contributing to the progression of glaucoma [12,13]. Thus, under glaucomatous conditions, effectively removing damaged mitochondria is an essential step for maintaining mitochondrial quality control and, consequently, protecting RGCs.

In mammalian cells, intracellular AIBP is localized in the mitochondria [117]. However, its roles in the mitochondria have yet to be extensively studied. Recently, we found for the first time the binding of AIBP to Parkin (PARK2) and mitofusin (MFN) 1 and 2, which are essential for regulating mitophagy and mitochondrial quality control, resulting in augmentation of mitophagy and protecting macrophages against apoptotic cell death in the context of atherosclerosis [20]. This anti-atherogenic effect of mitochondrial AIBP in the regulation of mitophagy was confirmed by others [118]. More importantly, elevated IOP significantly induced a reduction of retinal AIBP expression in mitochondria, suggesting that alterations in retinal AIBP expression are associated with the progression of glaucoma [12]. In parallel, AIBP deficiency decreased mitochondrial dynamics-related proteins, such as optic atrophy type 1 (OPA1) and dynamin-related protein 1 (DRP1), as well as mitochondrial oxidative phosphorylation (OXPHOS) complex proteins in the retina of *Apoa1bp^−/−^* mice [12].

Moreover, employing state-of-the-art technology, such as serial block-face scanning electron microscopy and three-dimensional electron microscopy tomography, it is well-defined that AIBP deficiency triggers mitochondrial dysfunctions in both Müller glia and RGCs in naïve *Apoa1bp^−/−^* mice [12]. To provide a detailed insight, mitochondria from Müller glia in naïve *Apoa1bp^−/−^* mice showed mitochondria fragmentation and ring-shaped mitochondria, which is a hallmark of mitochondrial stress, as well as depletion of mitochondrial cristae, resulting in the reduction of ATP production [12]. In parallel, AIBP deficiency causes mitochondrial abnormalities in RGC somas, such as mitochondrial fragmentation, cristae depletion, and ATP reduction. These findings importantly suggested that mitochondrial AIBP is a critical factor in maintaining mitochondrial network and function, and that pathological loss of retinal AIBP expression might exacerbate RGC degeneration and visual dysfunction (Figure 2).

Due to their low cholesterol content, mitochondria are highly linked to the progression of various diseases, including Alzheimer’s disease, Niemann–Pick Type C deficiency, and fatty liver disease. In particular, these conditions are characterized by increasing oxidative stress and compromised OXPHOS system [119,120]. While a novel role of mitochondrial AIBP has been identified in chronic inflammatory diseases [12,20,118], there is currently no evidence of exploring connection between AIBP and cholesterol within mitochondria. Therefore, the precise molecular mechanisms by which AIBP impacts cholesterol metabolism within mitochondria or how AIBP-mediated lipid raft dynamics in the plasma membrane affect mitochondrial quality control under various pathological conditions in glaucomatous neuroinflammation remains to be explored (Figure 2).

Our recent article has reviewed the adverse role of mitochondrial damage-associated molecular patterns (mtDAMPs) in glaucomatous neurodegeneration [88]. Mitochondria play a crucial role in maintaining homeostasis through various pathways, such as quality control protease, unfolded protein response, apoptotic cell death, and mitophagy. Under pathological conditions, cellular defense systems, including mitophagy, are overwhelmed due to sustained cellular stress and tissue damage. This leads to the release of mtDAMPs into either the intracellular or extracellular compartment [5]. Disrupted mitochondrial membrane potential (MMP) leads to the release of mtDAMPs, such as mtDNA, mtROS, cardiolipin, the mitochondrial transcription factor A, and formyl-peptides [121]. Several studies have demonstrated a link between MMP depolarization and glaucoma. In a rat model of glaucoma induced by chronic ocular hypertension, with the injection of cross-linking hydrogel into the anterior chamber MMP depolarization was prominent in RGCs [122]. In addition, elevated hydrostatic pressures triggered MMP depolarization in the neuronal cell culture system [123].

The release of mtDAMPs activates glial cells through PRRs of the innate immune responses, including the NLRP3 inflammasome, TLR9, and cGAS-STING-mediated signaling. Subsequent activation of PRRs increases type 1 interferon (IFN) expression and the levels of proinflammatory cytokines [121]. Specifically, NLRP3 inflammasome activity was upregulated in the retina of glaucomatous DBA/2J mice [124]. TLR9 mRNA expression was significantly increased in trabecular meshwork from patients with POAG and in vitro TM culture systems [125]. In addition, cGAS-STING activity was increased in RGCs following ischemia/reperfusion and in poly(dA:dT)-treated BV-2 microglial cells [126]. Given that RGCs are more vulnerable than glial cells to glaucomatous insult-induced mitochondrial stress [127,128] and that a defect in mitophagy in RGCs is linked to AIBP deficiency [20,118], released mtDAMPs from the *Apoa1bp^−/−^* RGCs could be the case that AIBP deficiency clearly linked to mitochondrial dysfunction and TLR4-mediated neuroinflammation via glial cell activation in glaucoma. Hence, it would be useful to validate and quantify the levels of mtDAMPs in glaucomatous human and mouse retinas (Figure 2).

## 5. AIBP-Mediated Neuroprotection in Glaucomatous Neuroinflammation and Neurodegeneration

AIBP protects RGCs against neuroinflammation and mitochondrial dysfunction in elevated IOP-induced retinal neurodegeneration [12]. Specifically, administration of recombinant AIBP enhanced RGC survival against apoptotic cell death and prevented inflammatory responses and cytokine production triggered by the activation of Müller glia, induced by acute IOP elevation in vivo [12]. Because the absence of AIBP remarkably induced the alterations of mitochondrial structure and function in Müller glia and RGCs in vivo [12], these findings importantly indicate that AIBP has a potential to modulate inflammatory mechanisms, protecting against mitochondrial dysfunction that is linked to Müller glia activation and RGC death in retinal neurodegeneration.

Remarkably, our emerging evidence demonstrated that restoring AIBP expression in the retina promotes RGC survival in experimental glaucoma [13]. In the study, we have identified AIBP as the protein regulating several mechanisms of glaucomatous neurodegeneration, including TLR4-mediated inflammatory signaling, excessive cholesterol accumulation, oxidative stress, mitochondrial dysfunction, and metabolic stress. Using a single intravitreal injection of adeno-associated virus (AAV)-AIBP, we found that restoring AIBP expression protected RGCs and their axons in several mouse models of glaucoma, including DBA/2J mice, microbead-induced ocular hypertension, and ONC [13]. In parallel, restoring AIBP expression improved visual function by enhancing RGC survival, evident in the increased amplitude of pattern electroretinogram, and enhanced visual acuity through increased spatial frequency against ONC injury. In addition, restoring AIBP expression decreased cholesterol accumulation, TLR4 and IL-1β expression, localization of TLR4 to lipid rafts, and inhibited AMP-activated protein kinase (AMPK) activation in the retina of glaucomatous DBA/2J mice [13]. Hence, our findings suggest for the first time that AIBP attenuates mitochondrial stress and neuroinflammation and protects RGCs by preventing Müller glia activation in glaucoma. We propose that reversing AIBP deficiency by AAV-AIBP delivery has the therapeutic potential to treat glaucoma.

## 6. Potential Role of AIBP in the NAD(P)HX Repair System of the Retina

To eliminate dangerous toxic metabolites, organisms have developed damage-control systems to remove or recycle toxic metabolites [129]. Nicotinamide adenine dinucleotide (reduced version, NADH; oxidized version, NAD^+^) and nicotinamide adenine dinucleotide phosphate (reduced version, NADPH; oxidized version, NADP^+^) are important cofactors for redox equivalents as well as non-redox NAD^+^-dependent enzymes to maintain metabolic homeostasis [130]. The NAD(P)HX repair system is a highly conserved cellular metabolic reaction, from prokaryotes to eukaryotes. The NAD(P)HX dehydratase and epimerase that catalyzes the repair of NAD(P)HX are widely expressed in all tissues [131]. The *APOA1BP* gene is also known as NAD(P)HX epimerase (NAXE) due to facilitating the interconversion of *R*-NAD(P)HX and *S*-NAD(P)HX epimers [132,133]. NAD(P)H is converted into NAD(P)HX by either glyceraldehyde-3-phosphate dehydrogenase (GAPDH) or spontaneously in an acidic pH or at high temperatures [134,135]. The converted NAD(P)HX exists in the *R*- and *S*-forms, which are thought to be toxic. AIBP, as an epimerase, converts the *R*-NAD(P)HX epimers to the *S*-form. Subsequently, the NAD(P)HX dehydratase converts the *S*-form back to NAD(P)H, resulting in the inhibition of cyclic NAD(P)HX production, which is an aberrant metabolite, inducing mitochondrial dysfunction. Given that the pathological implications of the NAD(P)HX repair system have recently received attention in human studies, until now, it has been reported a total of 30 clinical cases with 19 rare variants in AIBP that cause behavioral changes and lethal neurometabolic disorders, including ataxia, ophthalmoplegia, muscle weakness, and the regression of motor and cognitive functions [131]. In brief, these pathological variants in patients including 12 missense mutations (c.281C>A, c.255A>T, c.361G>A, c.368A>T, c.386G>C, c.565G>A, c.640A>G, c.641T>G, c.652G>A, c.653A>T, c.733A>C, and c.757G>A), 2 splicing mutations (c.516+1G>A and c.665-1G>A), 2 nonsense mutations (c.177C>A and c.196C>T), 2 frameshift mutations (c.743delC and c.743delCA), and 1 deletion/insertion mutation (c.804_807delinsA) can cause a neurometabolic condition and neuroinflammation in humans [131]. AIBP is highly conserved from human to mouse, with 84% DNA sequence homology and 88% protein sequence homology, suggesting the functional similarity between the two species. Nevertheless, the deficiency of AIBP in mice does not impact their growth and viability [16,136], indicating that a mouse may exhibit greater tolerance to toxic metabolites or that human conditions may have confounding genetic components [113].

Recent studies uncovered the neuroprotective function of nicotinamide (NAM) in both mouse models of glaucoma and patients with glaucoma [137,138]. However, there is no reported evidence that deficiency in the NAD(P)HX repair system contributes to the onset of glaucoma. However, in both mouse models of glaucoma and the patients with glaucoma, NAD or NAM levels are significantly lower than in control subjects [137,139]. NAM is a precursor form of NAD that is generated from vitamin B-3, or the precursor NAM, nicotinamide riboside, nicotinic acid, or nicotinamide mononucleotide (NMN), defined as the salvage pathway [140]. Therefore, metabolic boosting of NAD^+^ levels may have a beneficial effect on glaucoma treatment.

AIBP protein expression, but not mRNA expression, was reduced in RGCs in both a mouse model of glaucoma, DBA/2J mice, and patients with glaucoma [12,13]. Moreover, AIBP deficiency significantly reduced visual acuity measured by decreasing spatial frequency in male and female naïve *Apoa1bp^−/−^* mice [12]. Recent findings demonstrated that the plasma levels of NAD in 3-month-old *Apoa1bp^−/−^* mice tended to decrease compared to control mice, although the difference was not significant [141]. Since the retinal NAD^+^ or NAM levels have not been tested in *Apoa1bp^−/−^* mice [16,136], whether AIBP deficiency confers metabolic alterations in the retina is unclear. Nevertheless, a potential possibility is that, in part, accumulation of toxic metabolites, e.g., cyclic NAD(P)HX, in the retina of *Apoa1bp^−/−^* mice may have deleterious effects on metabolic and mitochondrial homeostasis, leading to RGC degeneration and subsequent vision loss (Figure 3). This hypothesis requires experimental testing.

## 7. Conclusions and Future Directions

We have reviewed the proposed functions of AIBP in protecting against the development of glaucoma, from inhibition of TLR-mediated inflammation via disruption of TLR4-associated lipid rafts to the regulation of mitochondrial function to NAXE activity. We have also reported reduced AIBP protein expression in glaucomatous human retina [13] and the retinas in mouse models of glaucoma [12,13]. However, there is no indication in the literature that there is a genetic link between *APOA1BP* variants and the risk of development of glaucoma in human populations. Additional genetic studies are required to evaluate this possibility.

In this article, we have emphasized two main concepts of AIBP actions beyond the NAXE-related repair systems. The first concept is that extracellular AIBP selectively reduces pathological lipid raft content in TLR4-expressing retinal glial cells, inhibiting TLR4-mediated inflammation and mitochondrial dysfunction in glaucoma. Here, we focused on the effect of AIBP on TLR4-associated lipid raft formation in glaucoma. However, we do not exclude a potential role of AIBP in the regulation of other TLRs localized to lipid rafts, including TLR2 and TLR3, because their expression is increased in human glaucomatous retina [142]. The second concept is the role of AIBP in mitochondria. As a result of the reduction of mitochondrial AIBP expression, RGC survival was decreased, and visual acuity was reduced in response to elevated IOP [12]. While the majority of reported research findings are to explain the role of AIBP in the regulation of cholesterol metabolism, future studies need to be conducted to test the requirement of cholesterol for the function of mitochondrial AIBP.

From the therapeutic perspective, glaucoma is closely associated with neuroinflammation that could be initiated by inflammatory receptor-lipid raft complex, thereby triggering oxidative stress and mitochondrial dysfunction. Because reduced AIBP expression in the retina is linked to glaucomatous neuroinflammation and mitochondrial dysfunction [12], sustained expression of AIBP, which promotes selective removal of excess cholesterol from TLR4-occupied lipid rafts in inflammatory retinal cells, could become a potential treatment to reduce neuroinflammation and mitochondrial dysfunction in human glaucoma. Clinical development of gene therapy using AAV in ocular disease has drawn attention because of its long-term stable transgene expression at levels that are therapeutic, even after a single injection. Hence, restoring AIBP expression by a single intravitreal injection of AAV-AIBP could be a potential treatment for glaucoma.

Here, we highlighted AIBP as a novel safeguard that protects against retinal neuroinflammation and mitochondrial dysfunction. Notably, the remarkable selectivity of AIBP toward TLR4-associated lipid rafts and regulation of mitochondrial dynamics and function may contribute to developing a new therapeutic strategy for the treatment of glaucoma.

## Figures and Tables

**Figure 1 cells-13-00198-f001:**
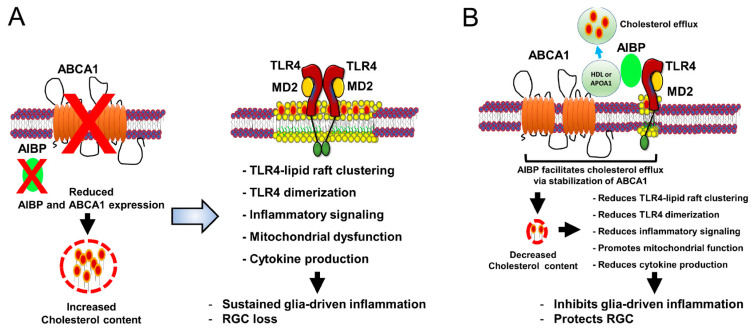
AIBP-mediated cholesterol efflux in glaucoma. Schematic overview of AIBP-associated protective effect on glia-driven neuroinflammation in glaucomatous retina. (**A**) Increased cholesterol accumulation by reducing AIBP and ABCA1 expression leads to TLR4-lipid raft clustering, TLR4 dimerization, TLR4-dependent inflammatory signaling, mitochondrial dysfunction, and cytokine production, resulting in sustained glia-driven neuroinflammation and RGC loss in glaucomatous retina. (**B**) Enhanced cholesterol efflux by AIBP to HDL or lipid-free APOA1 from TLR4-associated lipid rafts in activated glial cells reduces cholesterol accumulation and the ensuing inhibition of TLR4-dependent inflammation. This results in the inhibition of glia-driven neuroinflammation and protects RGC against cell death in the retina.

**Figure 2 cells-13-00198-f002:**
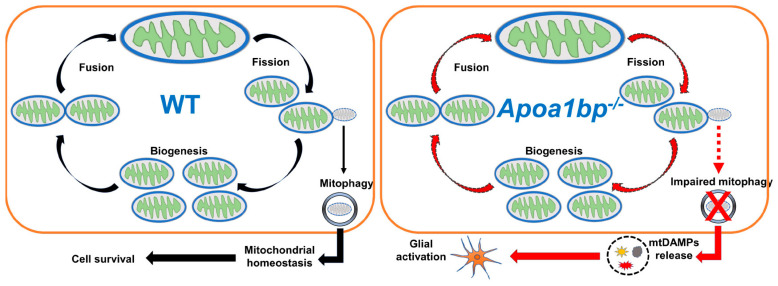
Mitochondrial AIBP in glaucoma. The role of mitochondrial AIBP in glaucoma. Impairment of mitochondrial function and defective mitophagy in *Apoa1bp^−/−^* retina may contribute to glial cell activation via mitochondrial DAMPs (mtDAMPs) that are released from apoptotic cells in glaucoma.

**Figure 3 cells-13-00198-f003:**
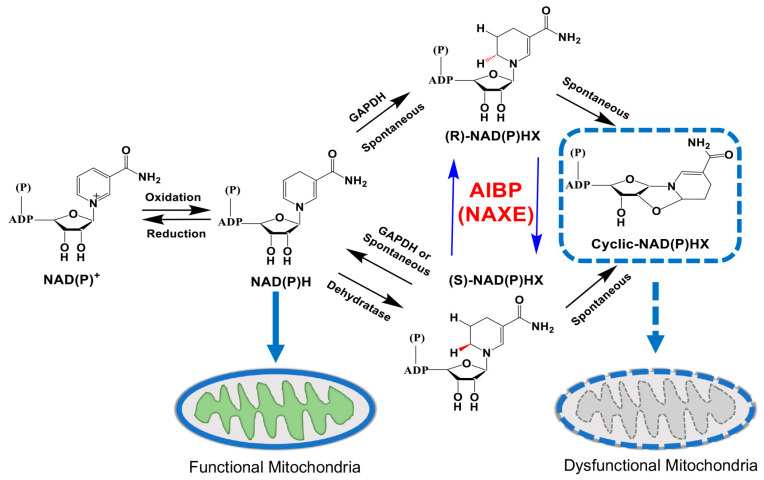
The NAXE-associated NAD(P)HX repair system. The NAD(P)HX repair system. Nicotinamide adenine dinucleotide (NAD^+^, oxidized form: NADH) and Nicotinamide adenine dinucleotide phosphate (NADP^+^, oxidized form: NADPH) are hydrated by the glyceraldehyde 3-phosphate dehydrogenase (GAPDH) or spontaneously to form *R*- and *S*-epimers. The *R*-form of NAD(P)HX is converted into the *S*-form of NAD(P)HX by AIBP (NAXE) and the *S*-form is then catalyzed by the ATP-dependent dehydratase. AIBP deficiency leads to the accumulation of cyclic NAD(P)HX, a toxic inhibitor of cellular NADH dehydrogenases, in an irreversible path, which in turn decreases the activity of mitochondrial OXPHOS complex I (C-I) and pyruvate dehydrogenase complex (PDHc).

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
