# Peer review of "AIBP: A New Safeguard against Glaucomatous Neuroinflammation"

_cells, 2024, doi:10.3390/cells13020198_

Round 1

Reviewer 1 Report

Comments and Suggestions for Authors

The article, as I understood, is a literature review, if it is correct, should be clear in the title. I suggest to change the title like: 

A New Safeguard against Glaucomatous 2 Neuroinflammation: literature review report on AIBP

Anyway into the text you wrote "Because we have revealed a reduction of retinal 440 AIBP expression in both glaucomatous human and mouse retinas, it is crucial to investi- 441 gate the relevance of AIBP in human glaucoma, particularly considering the absence of 442 genetic variation in the APOA1BP gene. 4" is it one of Your research ? If YES please rewrite the article (it is very nice article) with your experience and your research. Dividing te article in Material and methods (describe your research)

Results (reporting your results)

Discussion (compare with literature your results)

Conclusions (should be leave like it is )

The article is very interesting

Author Response

Reviewer 1 Comments:

  1. The article, as I understood, is a literature review, if it is correct, should be clear in the title. I suggest to change the title like: A New Safeguard against Glaucomatous Neuroinflammation: literature review report on AIBP

Response : We appreciate the suggestion, but we would like to keep the title as it is. We mentioned in the abstract and this is this is a review article. We added the following sentence “In this article, we review current evidence for the emerging role of apolipoprotein A-I binding protein (AIBP) as an important anti-inflammatory and neuroprotective factor in the retina.” in the abstract.

  1. Anyway into the text you wrote "Because we have revealed a reduction of retinal 440 AIBP expression in both glaucomatous human and mouse retinas, it is crucial to investi- 441 gate the relevance of AIBP in human glaucoma, particularly considering the absence of 442 genetic variation in the APOA1BP gene. 4" is it one of Your research? If YES please rewrite the article (it is very nice article) with your experience and your research. Dividing te article in Material and methods (describe your research), Results (reporting your results), Discussion (compare with literature your results), Conclusions (should be leave like it is ). The article is very interesting.

Response : Thank you for having interest in our research. Yes, these findings are from our studies. In previous studies, we demonstrated that the AIBP protein expression was decreased in human and mouse glaucomatous retinas. These findings were published in Redox Bio (1) and bioRxiv (2) and support our hypothesis that a reduction of retinal AIBP expression contributes to the development of glaucoma in humans. Because AIBP (also known as NAXE) acts not only as an epimerase in the NAD(P)HX repair system but also as a modulator of cholesterol metabolism and mitochondria function, AIBP loss-of-function variants cause lethal neurometabolic diseases as responded in Response #1 for the Reviewer #2. However, there is no report for SNPs or gene variants in AIBP that are associated with human glaucoma. Therefore, further examination of AIBP loss-of-function variants in patients with glaucoma will bring a better idea to understand mechanism underlying AIBP contribution to human glaucoma.

Reviewer 2 Report

Comments and Suggestions for Authors

The authors reviewed the literature regarding AIBP (NAXE) as a new safeguard against glaucomatous neuroinflammation and the authors appear conducted many studies on the AIBP (NAXE). I have some comments that need to be addressed before the manuscript can be considered for publication in this journal:

1.    IF there is any gene mutation or SNPs in AIBP that associated with human disease condition, please give a brief discussion.

2.    Is AIBP gene highly conserved between human and mouse? If so, the authors can mention it in the manuscript. For example, some genes are not conserved, mutations in myocilin gene are associated with human glaucoma, however, myocilin gene KO mice did not induce glaucoma and the mouse needs to carry a human myocilin gene to induce glaucoma.

3.    Do neurons such as retinal ganglion cells (RGC) express and secrete AIBP? Do RGC synthesize cholesterol? If so, does AIBP in RGC also exert its protective role in glaucoma via down regulation of neuroinflamation? Does cholesterol in RGC play a role in RGC death?

4.    From line 225, it appears there is a typo, “a extensively studied” should be “an extensively studies).

5.    What is the half life of AIBP?  From line 359, the authors may mention which technique (AAV or lentivirus carrying AIBP, or fusion protein with cell penetration peptide, etc) was used to restore AIBP expression, also for AAV-AIBP, is there a special serotype of AAV perform better than others in retina? For example, some teams use AAV2, some teams use other serotypes of AAV.

6.    The authors used “NADPH” from line 380, and “NAD(P)H” from line 387, what is the difference between “NADPH” and “NAD(P)H”.

7.    From line 410-411, the authors stated that “AIBP expression was predominately reduced  in …”, please add “mRNA or protein expression  or both”, also make sure the AIBP level was reduced in human patients with glaucoma. Because mRNA expression is not always correlated with protein expression.

8.    For reference 91, it appears the authors missed the journal name and page numbers. “Plos one” …..???

9.    There are many abbreviations, the authors may add “Abbreviations or Synonyms” section in the end.

Thank you!

Author Response

Reviewer 2 Comments:

The authors reviewed the literature regarding AIBP (NAXE) as a new safeguard against glaucomatous neuroinflammation and the authors appear conducted many studies on the AIBP (NAXE). I have some comments that need to be addressed before the manuscript can be considered for publication in this journal:

  1. IF there is any gene mutation or SNPs in AIBP that associated with human disease condition, please give a brief discussion.

Response : Thank you for suggestion. Now, we added the following text to section 6.

In brief, these pathological variants in patients including 12 missense mutations (c.281C>A, c.255A>T, c.361G>A, c.368A>T, c.386G>C, c.565G>A, c.640A>G, c.641T>G, c.652G>A, c.653A>T, c.733A>C, and c.757G>A), 2 splicing mutations (c.516+1G>A and c.665-1G>A), 2 nonsense mutations (c.177C>A and c.196C>T), 2 frameshift mutations (c.743delC and c.743delCA), and 1 deletion/insertion mutation (c.804_807delinsA) can cause the abnormal neurometabolic condition and neuroinflammation in human.”

  1. Is AIBP gene highly conserved between human and mouse? If so, the authors can mention it in the manuscript. For example, some genes are not conserved, mutations in myocilin gene are associated with human glaucoma, however, myocilin gene KO mice did not induce glaucoma and the mouse needs to carry a human myocilin gene to induce glaucoma.

Response : Yes, AIBP is highly conserved between human and mouse. We now added this text to section 6: “AIBP is highly conserved from human to mouse that shows 84% DNA sequence homology and 88% protein sequence homology, suggesting the functional similarity between two species.”

  1. Do neurons such as retinal ganglion cells (RGC) express and secrete AIBP? Do RGC synthesize cholesterol? If so, does AIBP in RGC also exert its protective role in glaucoma via down regulation of neuroinflammation? Does cholesterol in RGC play a role in RGC death?

Response : These are very important questions. Based on our studies, AIBP is predominantly expressed in RGC (1 and 2). Since AIBP is known to be secreted, we hypothesized that the secreted AIBP protein from RGC could protect RGC against glial-mediated neuroinflammation. In the retina, Müller glia and RPE are main cell types for cholesterol synthesis and transfer it to RGC (3-5). RGC is a major cholesterol consumer in the retina and exclusively expresses cytochrome P450 46A1 (CYP46A1) which can remove excess cholesterol provided by Müller glia and RPE from RGC. We do not have the answer for “Do RGC synthesize cholesterol?” and “Does cholesterol in RGC play a role in RGC death?” at this moment. We will further study to understand the role of AIBP and cholesterol metabolism in RGC viability.

  1. From line 225, it appears there is a typo, “a extensively studied” should be “anextensively studies).

Response : We corrected it.

  1. What is the half life of AIBP?  From line 359, the authors may mention which technique (AAV or lentivirus carrying AIBP, or fusion protein with cell penetration peptide, etc) was used to restore AIBP expression, also for AAV-AIBP, is there a special serotype of AAV perform better than others in retina? For example, some teams use AAV2, some teams use other serotypes of AAV.

Response : In a 2023’s study, we used AAV-AIBP to test in vivo efficacy of AIBP in a mouse model of glaucoma and found sustained AIBP expression for 5 month (2). In the same study, we used AAV DJ/8 serotype which closely mimic AAV8 and AAV9 serotype (6) to enhance AAV-AIBP transduction to CNS. However, AAV2 serotype is widely used to transduce genes to the retina.

  1. The authors used “NADPH” from line 380, and “NAD(P)H” from line 387, what is the difference between “NADPH” and “NAD(P)H”.

Response : NADPH includes only nicotinamide adenine dinucleotide phosphate and NAD(P)H includes both nicotinamide adenine dinucleotide (NADH) and nicotinamide adenine dinucleotide phosphate (NADPH).

  1. From line 410-411, the authors stated that “AIBP expression was predominately reduced  in …”, please add “mRNA or protein expression  or both”, also make sure the AIBP level was reduced in human patients with glaucoma. Because mRNA expression is not always correlated with protein expression.

Response : In a 2020’s study, we demonstrated that mRNA level of Apoa1bp gene was not changed while the AIBP protein expression was reduced (1). We revised as below.

“AIBP protein expression, but not mRNA expression, was predominantly reduced in RGCs in both a mouse model of glaucoma”

  1. For reference 91, it appears the authors missed the journal name and page numbers. “Plos one” …..???

Response : Thank you for checking with this. We corrected it.

  1. There are many abbreviations, the authors may add “Abbreviations or Synonyms” section in the end.

Response : We added “Abbreviations” to the manuscript

REFERENCES cited in response to Reviewers

  1. Choi, S.-H.; Kim, K.-Y.; Perkins, G.A.; Phan, S.; Edwards, G.; Xia, Y.; Kim, J.; Skowronska-Krawczyk, D.; Weinreb, R.N.; Ellisman, M.H.J.R.b. AIBP protects retinal ganglion cells against neuroinflammation and mitochondrial dysfunction in glaucomatous neurodegeneration. 2020, 37, 101703.
  2. Ju, W.K.; Ha, Y.; Choi, S.; Kim, K.Y.; Bastola, T.; Kim, J.; Weinreb, R.N.; Zhang, W.; Miller, Y.I.; Choi, S.H. Restoring AIBP expression in the retina provides neuroprotection in glaucoma. bioRxiv 2023, doi:10.1101/2023.10.16.562633.
  3. Gabrielle, P.H. Lipid metabolism and retinal diseases. Acta Ophthalmol 2022, 100 Suppl 269, 3-43, doi:10.1111/aos.15226.
  4. Chen, M.; Luo, C.; Zhao, J.; Devarajan, G.; Xu, H. Immune regulation in the aging retina. Prog Retin Eye Res 2019, 69, 159-172, doi:10.1016/j.preteyeres.2018.10.003.
  5. Is 24(S)-hydroxycholesterol a potent modulator of cholesterol metabolism in Müller cells? An in vitro study about neuron to glia communication in the retina. Exp Eye Res. 2019 Dec:189:107857. doi: 10.1016/j.exer.2019.107857. Epub 2019 Oct 22.
  6. Grimm D, Lee JS, Wang L, Desai T, Akache B, Storm TA, Kay MA. 2008. In vitro and in vivo gene therapy vector evolution via multispecies interbreeding and retargeting of adeno-associated viruses. J Virol82:5887–5911. doi: 10.1128/JVI.00254-08.

Round 2

Reviewer 1 Report

Comments and Suggestions for Authors

If I will be a reader of you article, I'd like to understand immediately from the title if your article is a review or not, anyway if you don't want to change the title one word "review" should be added. Anyway the Editor will decide. 

Author Response

We appreciate the suggestion again and have revised the text as follows, “In this review article, we summarize current evidence for the emerging role of apolipoprotein A-I binding protein (AIBP) as an important anti-inflammatory and neuroprotective factor in the retina.” in the abstract. Also, the review article notice on the initial page of the revised manuscript draft makes it unnecessary to include it in the title. 

Thank you for your consideration